REGISTERED REPORT PROTOCOL

# Effects of mild-to-moderate sensorineural hearing loss and signal amplification on vocal emotion recognition in middle-aged–older individuals

**Mattias Ekberg**[1]*, **Josefine Andin**[1], **Stefan Stenfelt**[2], **Örjan Dahlström**[1]

**1** Swedish Institute for Disability Research, Department of Behavioural Sciences and Learning, Linköping University, Linköping, Sweden, **2** Department of Biomedical and Clinical Sciences, Linköping University, Linköping, Sweden

* henrik.mattias.ekberg@liu.se

This is a Registered Report and may have an associated publication; please check the article page on the journal site for any related articles.

## Abstract

Previous research has shown deficits in vocal emotion recognition in sub-populations of individuals with hearing loss, making this a high priority research topic. However, previous research has only examined vocal emotion recognition using verbal material, in which emotions are expressed through emotional prosody. There is evidence that older individuals with hearing loss suffer from deficits in general prosody recognition, not specific to emotional prosody. No study has examined the recognition of non-verbal vocalization, which constitutes another important source for the vocal communication of emotions. It might be the case that individuals with hearing loss have specific difficulties in recognizing emotions expressed through prosody in speech, but not non-verbal vocalizations. We aim to examine whether vocal emotion recognition difficulties in middle- aged-to older individuals with sensorineural mild-moderate hearing loss are better explained by deficits in vocal emotion recognition specifically, or deficits in prosody recognition generally by including both sentences and non-verbal expressions. Furthermore a, some of the studies which have concluded that individuals with mild-moderate hearing loss have deficits in vocal emotion recognition ability have also found that the use of hearing aids does not improve recognition accuracy in this group. We aim to examine the effects of linear amplification and audibility on the recognition of different emotions expressed both verbally and non-verbally. Besides examining accuracy for different emotions we will also look at patterns of confusion (which specific emotions are mistaken for other specific emotion and at which rates) during both amplified and non-amplified listening, and we will analyze all material acoustically and relate the acoustic content to performance. Together these analyses will provide clues to effects of amplification on the perception of different emotions. For these purposes, a total of 70 middle-aged-older individuals, half with mild-moderate hearing loss and half with normal hearing will perform a computerized forced-choice vocal emotion recognition task with and without amplification.

**Data Availability Statement:** This is a stage one registered report and we have not yet begun data collection there are no data to be made available. Data will be stored in a public repository after finished data collection, and a link (URL) will be made publicly available upon publication of the stage two manuscript (full article).

**Funding:** The author(s) received no specific funding for this work.

**Competing interests:** The authors have declared that no competing interests exist.

# 1. Introduction

Hearing loss is among the leading causes of disability globally with an increasing prevalence with age. It leads to difficulties in communication which can contribute to social isolation and diminished well-being [1]. In Sweden, it has been estimated that approximately 18% of the population have a hearing loss. While the prevalence is comparatively much higher in the oldest age groups (44.6% in the age group of 75–84 and 56.5% in the age group of 85+ years) the prevalence in the age groups of 55–64 and 65–74 (middle-age-to-old) is also relatively high (24.2 and 32.6%) [2]. This shows that hearing loss is not only a concern for the oldest in the population. Because of the importance of emotion recognition for interpersonal interaction and, thus, indirectly to measurements of well-being such as perceived quality of life and the presence/absence of depression, the effects of hearing loss, amplification, and auditory rehabilitation on the recognition and experience of vocal emotions, have been identified as important research topics [3]. In the present study, we will investigate how having a mild to moderate hearing loss and using hearing aids with linear amplification influences vocal emotion recognition for verbal and non-verbal materials.

## Vocal emotion recognition

Emotions can be defined as "episodic, relatively short-term, biologically-based patterns of perception, experience, physiology, action and communication that occur in response to specific physical and social challenges and opportunities" ([4], p. 468). Perceiving and recognizing emotions are important and deficits in emotion recognition can have negative effects on interpersonal relationships and different social contexts such as job environments [3]. Vocal emotion expressions are one source through which we perceive others' emotions. Vocal emotion recognition depends on a complex interplay between different factors (see [5] for a review of several such factors). Both the producer and observer of an emotion expression are influenced by their psychobiological architecture (e.g. the structure and function of the auditory system) and neurophysiological mechanisms, as well as by context and codes for expressing and interpreting emotion expressions. In the receiver, attributions concerning the emotional state of the expressions are made based upon schematic recognition and inference rules which are influenced by sociocultural context [5]. Schemata can be understood as cognitive organizing systems which control information simplification to manage its processing. These schemata can guide perception by means of providing categories, for example different emotions, and summarizing the most common and distinguishing attributes of different categories [6]. The expresser produces distal cues, which in the case of vocal expressions, are sounds with particular acoustic properties. These distal cues are transmitted to the perceiver and transformed into proximal cues; perceived voice features on which inferences are made [5]. In support of such a model, Bänziger Hosoya and Scherer have shown that the recognition of speech-prosody embedded emotion expressions is predicted by combinations of perceived voice features, which in turn are best explained as products of different complex combinations of acoustic features such as intensity, fundamental frequency, and speech-rate [7]. However, neither acoustic features nor perceived voice features fully predict vocal emotion recognition in normal hearing individuals [6], which indicates that recognition also depends upon other factors, for example differences in schematic knowledge of emotion expressions. Nevertheless, analysis of acoustic features may provide cues for understanding why some emotion expressions are more easily recognized than others, and why particular emotions are commonly mistaken (see for example [8], for such an analysis). Therefore, in the present study, we will also examine acoustic features of different emotional expressions and relate these features to vocal emotion recognition.

Emotions are vocally expressed through emotional prosody, as well as through non-verbal vocalizations [9]. Chin, Bergeson, and Phan define prosody as "the suprasegmental features of speech that are conveyed by the parameters of fundamental frequency, intensity, and duration" ([10], p.356). Non-verbal vocalizations, or affect bursts, are brief non-linguistic sounds such as for example laughter, to express happiness or amusement, and crying, to express sadness [9]. Prosody, including emotional prosody, evolves relatively slow in speech, requiring temporal tracking and integration over time in the perceiver [11]. In the emotion expresser, patterns of emotional prosody are likely affected to a high degree by socioculturally grounded norms of communication, while the expressive patterns of non-verbal vocalizations are driven, involuntarily, by psychobiological changes in the expresser [4]. Generally, non-verbal vocalizations are recognized more accurately than emotional speech prosody [5, 12], and recognition accuracy for different emotions varies between emotional prosody in speech and non-verbal vocalizations [12–15]. Research on how well different emotions of both stimulus types are recognized compared to other emotions has so far not yielded consistent results (see for example [4] and [15]). However, with regard to emotional prosody specifically, Scherer [5] and Castiajo and Pinheiro [16] argue that anger and sadness are more accurately recognized compared to fear, surprise, and happiness. Anger, fear, sadness, happiness, and interest, expressed through prosody are not commonly confused for other emotions [5]. It is more common, however, for emotions which can be considered as variants of a broader category to be confused for one-another, such as for example pleasure and amusement within the category of happiness [5]. With regard to non-verbal vocalizations, Lima et al. showed that normal hearing younger adults could rapidly recognize expressions of eight different emotions with high accuracy, even under concurrent cognitive load, indicating that, at least for this population, the recognition of non-verbal vocalizations may depend on predominately automatic mechanisms [17]. The ability to accurately recognize non-verbal emotion expressions declines with aging [18], as does the ability to recognize emotional prosody in speech [19]. Amorim et al. suggest that this decline happens due to age-related changes in brain regions which are involved in the processing of emotional cues [18]. Hearing loss and aging both independently and conjointly contribute to a diminished ability to accurately recognize speech embedded vocal emotion expressions generally [19]. To our knowledge, however, no previous study has examined the effects of mild to moderate sensorineural hearing loss on non-verbal emotion expressions. Therefore, in the present study recognition of non-verbal expressions as well as emotional prosody will be investigated.

## Vocal emotion recognition under mild-to-moderate hearing loss

Damage to any part of the auditory system can result in hearing loss [20]. The most common form of hearing loss is cochlear hearing loss [21], a type of peripheral sensorineural hearing loss involving damage to the structures within the cochlea, most commonly to the outer hair cells (OHCs), but also to the inner hair cells (IHCs) [21–23]. Although not yet fully described, emotional expressions are characterized by different acoustic parameters. Processing and integration of these acoustic cues are, when correctly identified, helping the brain to recognize and differentiate between several emotions. Example of acoustic parameters are frequency perception, frequency discrimination, pitch and speech-related cues.

Consequences of sensorineural hearing loss include reduced sensitivity, dynamic range, and frequency selectivity [21]. Reduced sensitivity is the decreased ability to perceive sounds, often at certain frequencies. The loss of dynamic range entails that quiet sounds will be inaudible while loud sounds will be unaffected. Reduced frequency selectivity is the decreased ability to resolve spectral components of complex sounds [21], which is likely related to poorer pitch

perception [21]. The presence of hearing loss and its degree of severity is commonly determined by pure-tone audiometry in which the hearing threshold levels measured in hearing level (dB HL) are determined at frequencies between 125 and 8000 Hz [24]. The pure tone average (PTA) is the average hearing threshold at specified frequencies, and one often reported average is PTA4 including the hearing thresholds at 500, 1000, 2000, and 4000 Hz. The degrees of hearing loss are categorized based on hearing thresholds. There is no consensus about the exact categorization of degrees of hearing loss based on hearing thresholds. However, one common categorization, used by for example the World Health Organization (WHO) is to divide degrees of hearing loss into mild (PTA4 of 26 to 40 dB HL), moderate (PTA4 of 41 to 60 dB HL), severe (PTA4 of 61–80 dB HL) and profound (PTA4 of more than 81 dB HL) [24]. Mild-to-moderate hearing loss is most common, making up 92% of cases [1].

The most common intervention for people with hearing loss is the use of hearing aids [21]. However, many individuals with hearing loss do not have access to or do not use hearing aids [25]. Hearing aids using linear amplification can restore the audibility of quiet sounds but do not solve the problems of a loss of dynamic range and reduced frequency selectivity [19]. Modern hearing aids aim to restore the outer hair cells' (OHCs) function by non-linear amplification and compression of sounds, which involves the selective amplification of more quiet sounds, but do not fully restore the normal neural activity patterns in the ear and brain, and thus do not fully restore normal perception [26].

Several studies have found mild-to-moderate hearing loss to be related to deficits in vocal emotion recognition ability [19, 27, 28]), an ability that is not mitigated by hearing aid use [27], but also see [29, 30]. However, it has been suggested that what is interpreted as vocal emotion recognition deficits should rather be interpreted as general deficits in prosody recognition (including linguistic and non-linguistic prosody; [30, 31]). Concerning recognition of different vocal emotions, Christensen et al. found no significant interaction between emotion category and hearing loss [19]. Goy et al., however, found such an interaction, with sadness being recognized accurately significantly more often compared to anger, disgust, fear and happiness, but not compared to surprise or neutrality. In individuals with hearing loss, confusions between different vocal emotions appear to be much more common in comparison to normal hearing individuals [31].

**Aims.**   The overall aim is to deepen our understanding of the effects of hearing loss and signal amplification on vocal emotion recognition. It is unclear how mild-moderate hearing loss and linear amplification through a hearing aid effects the perception of auditory cues important for (a) being able to perceive and (b) being able to discriminate between emotions.

More specifically, by examining accuracy for different emotions (what emotions are easiest to accurately identify) and patterns of confusion (which emotions are mixed up when inaccurately identified), for individuals with normal hearing and for individuals with hearing loss (listening with and without linear amplification), and by comparing performance with acoustic analysis of sentences and non-verbal expressions with different emotional emphasis, the aims are to examine:

- Which acoustic parameters are important for detection of and discrimination between different emotions (by examination of performance of normal hearing participants)

- How the effect of hearing loss affects emotion recognition (by comparing performance of participants with and without hearing loss)

- How emotion recognition is affected by linear amplification (by examination of performance of participants with hearing loss using linear amplification)

Based on the literature, we predict that:

1. individuals with hearing loss will have poorer recognition compared to normal hearing individuals for emotions expressed verbally, regardless of acoustic features and regardless of the use of linear amplification, and for emotions expressed non-verbally when linear amplifications is not used

2. individuals with and without hearing loss will not differ in accuracy for non-verbal expressions when linear amplification is used

3. patterns of confusion will differ between individuals with and without hearing loss for both verbally and non-verbally expressed emotions

4. vocal emotions which are more distinct in terms of acoustic parameter measures, will be recognized with higher accuracy for both groups, but emotions that are distinguished mainly based on frequency parameters will be less correctly identified by the hearing loss group

5. the more salient the amplitude-related acoustic parameters are for an emotion, the better that emotion will be identified when linear amplification is used compared to not.

## 2. Methods

### 2.1. Participants

Using a 2 x 6 mixed design (group x emotion), 80% power, 5% significance level, correlation between repeated measures of 0.5 and with N = 56 participants (a reasonable number of participants from a recruitment point of view), we will be able to detect an effect as small as $\eta^2 = .02$ (f = 0.14); a next to small effect size. Power analysis was performed using G*Power version 3.1.9.7 [32].

Twenty-eight native Swedish speaking participants, aged 50–75, with mild-to-moderate, bilateral, symmetric sensorineural hearing loss (PTA4 of 30–60 dB HL), who have been using hearing aids for at least one year, and 28 age-matched native Swedish speaking participants with normal hearing will be recruited. Approximately equally number of men and women will be included in both groups. The choice of PTA4 of $\geq$30 dB as a criterion for the hearing loss group ensures that they benefit from hearing aid usage. The audiometric profiles of participants with hearing loss and information about which type of hearing aids they use, will be obtained through an audiological clinic, from participants who give their consent. The presence of normal hearing will be determined through audiometric testing with the criteria of equal or better than 20 dB HL at all frequencies between 125 and 4000 Hz and no worse than 30 dB HL at 8000Hz. Since an association between general cognitive ability (G) and emotion recognition ability have been established [33], the subtest Matrices from the Swedish version of WAIS-IV, which is strongly correlated with G, will be used for all participants [34]. Participants who perform below two standard deviations from the mean for their age span will be excluded from analysis. In addition, participants that have any of the following self-reported diagnoses or problems will be excluded from participation: hyperacusis, neurological disorders affecting the brain (e.g. multiple sclerosis or epilepsy), severe tinnitus (which is perceived to cause impairment and disability), developmental psychiatric disorders (e.g. ADHD, autism spectrum disorders or intellectual disability), mood and anxiety disorders (e.g. social anxiety disorder or depression), and the experience of great difficulties in identifying and describing one's own emotions. Before being invited to participate in the study, interested individuals will fill out an online questionnaire, including questions about health problems and diagnoses described above, educational attainment, age, gender, and native language. Those who fulfill

the inclusion criteria will be invited to Linköping University (Linköping, Sweden), where they will perform the experiment. All participants will sign a letter of informed consent. We will follow the declaration of Helsinki, and the project is approved by the Swedish Ethical Review Authority (Dnr: 2020–03674).

## 2.2. Task and study design

**Stimuli material.** The stimuli material is based on fourteen emotionally neutral sentences from the Swedish version of the hearing in noise test (HINT, [35]), and non-verbal emotion expressions. Four actors–an older female (69 years old), an older male (73 years old), a young female (19 years old) and a young male (29 years old)–were recorded when reading the sentences and when producing non-verbal expressions (sound expressions without using language), all with emotional prosody expressing different emotions of high and low intensity. The emotions included in the recordings are anger, happiness, sadness, fear, surprise. and interest For the sentences, prosodically neutral versions were also recorded. For the non-verbal expressions, the actors were instructed to imagine themselves experiencing the different emotions and to make expressions with sounds, without language which match those emotions. This entails some variation of the specific sounds expressed for specific emotions by different participants. The stimuli were recorded in a studio at the Audiology clinic at Linköping University Hospital, with the aid of a sound technician. Recordings were made in Audacity$^{TM}$ [36] using high-quality equipment, 24-bit resolution, and a 44.1 kHz sampling rate. From each actor, the clearest and most clean recordings out of two or more repetitions for each sentence and non-verbal vocalization was selected. With very few exceptions the sentences are approximately 2–3 seconds long and non-verbal expressions are 1–2 seconds long.

**Validation of stimulus material.** Sentences and non-verbal expressions are validated in an online administered task, where participants classify the perceived emotion by selecting from a list including the different emotions, and a neutral choice ("none of the described emotions/other emotion") to reduce bias in the responses. Only sentences and non-verbal expressions which a majority of participants (>50%) classify as expressing the emotion intended by the actor will be included in the experiment Preliminary results from the validation show that interest overall as a category is not sufficiently well recognized in our material. Interest will therefore not be included in the study.

**Study design.** Stimuli will be presented through headphones. In the aided listening conditions participants with hearing loss will use a master hearing aid system with linear amplification using the Cambridge formula for linear hearing aids [37] in which each stimulus is tailored to each participants audiogram during presentation. Participants will sit in front of a computer screen and will be presented with the written question "Which emotion was expressed in the recording you just heard?" with the options; happiness, anger, fear, sadness, surprise, and neutral for sentences, and the same categories excluding neutral for non-verbal expressions. The task is to identify the correct emotion by button-press. Two seconds after the participant's response, the next trial will be presented. The purpose of the lag between response and stimulus presentation is to allow for shifting of attention from responding to listening. Stimuli will be presented using PsychoPy version 3,0 (see [38] for a description of an earlier version).

Sentences and non-verbal expressions will be divided into separate sessions, and participants will have the opportunity to pause briefly between sessions. Participants with hearing loss will perform a total of four sessions; two sessions with hearing aids (aided listening condition), one for sentences and one for non-verbal expressions, and two sessions without hearing aids (unaided listening condition), one for sentences and one for non-verbal expressions. The

order of listening conditions will be balanced across participants. To have the same test-time and load, participants with normal hearing will perform all sessions, two with sentences as stimuli and two with non-verbal expressions, but only one, randomly chosen, of each will be included in the analyses. Stimuli within sessions will be presented in balanced order. Two different sets of sentences and non-verbal expressions will be used for comparison within each listening condition.

## 2.3. Analyses

**Acoustic analyses.** For each recording of verbal and non-verbal emotions the acoustic parameters of the Geneva Minimalistic Parameter Set (GeMAPS, [39]) will be extracted using the OpenSmile toolkit v.2.3 [40]. The GeMAPS consist of a set of acoustic parameters which have been proposed as a standard for different areas of automatic voice analysis, including the analysis of vocal emotions. The parameters of GeMAPS have been shown to be of value for analyzing emotions in speech in previous research [39] and are described in Table 1.

To control for baseline parameter differences between speakers' voices, raw values for each parameter will be centered on each speaker's neutral voice [8], while mean and standard deviations will allow for z-transformations of separate parameters within speaker and emotion. Acoustic parameters will be calculated for each recording separately. The mean and standard

**Table 1. A description of the acoustic parameters of the GeMAPS as discussed in Eyben et al. [39].**

| Parameters | Explanation |
| --- | --- |
| *Frequency related* | |
| Fundamental frequency (F0) | the logarithmic fundamental frequency, F0, on a semitone scale starting at 27.5 Hz. |
| Pitch (PT) | |
| Jitter | deviations in individual consecutive F0 period lengths the center frequency of the first formant. |
| Frequency–formant 1 | the center frequency of the first formant |
| Frequency–formant 2 | the center frequency of the second formant |
| Frequency–formant 3 | the center frequency of the third formant |
| *Energy/Amplitude/Intensity related* | |
| Shimmer | difference of the peak amplitudes of consecutive F0 periods |
| Loudness | an estimate of the perceived signal intensity from an auditory spectrum |
| Harmonics-to-noise ratio | relation of energy in harmonic components to energy in noise-like components |
| *Spectral (balance)-related components* | |
| Alpha ratio | ratio of the summed energy from 50–1000 and 1–5 kHz |
| Hammarberg index | ratio of the strongest energy peak in the 0–2 kHz region to the strongest energy peak in the 2–5 kHZ region |
| Spectral slope 0–500 Hz | linear regression slope of the logarithmic power spectrum within the given band |
| Spectral slope 500–1500 Hz | linear regression slope of the logarithmic power spectrum within the given band |
| Relative energy–formant 1 | the relative energy of the first formant and the ratio of the energy of the spectral harmonic peak at the first formant's center frequency to the energy of the spectral peak at the fundamental frequency |
| Relative energy–formant 2 | The relative energy of the second formant and the ratio of the energy of the spectral harmonic peak at the second formant's center frequency to the energy of the spectral peak at the fundamental frequency |
| Relative energy–formant 3 | The relative energy of the third formant and the ratio of the energy of the spectral harmonic peak at the third formant's center frequency to the energy of the spectral peak at the fundamental frequency |

deviation of the z-transformed values of specific parameters within separate emotions will for each emotion generate a coordinate in a multi-parameter-dimensional space (where the origin represents neutral). For illustrative purposes, a three-dimensional space where the axes represent weighted values of frequency, intensity and spectral balance related parameters will be outlined. For non-verbal expressions, the same procedure will be followed, with the exception that each emotion will be compared to the mean of all other emotions.

**Behavioural analyses.** First, comparisons between the normal hearing and the hearing loss group without amplification will be made using one 2 x 6 mixed ANOVA, for verbal stimuli, and one 2 x 5 mixed ANOVA, for non-verbal stimuli, all with group as the between-subjects variable and emotion as the within-subjects variable. To relate the effect of hearing loss on performance, differences in performance over groups and emotions will be described in relation to distance matrices and patterns of confusion from the acoustic analyses.

Second, effects of linear amplification will be analyzed in the hearing loss group by one 2 x 6 repeated measures ANOVA, for verbal stimuli, and one 2 x 5 repeated measures ANOVA, for non-verbal stimuli, both with listening condition (amplified and non-amplified) and emotion as factors. To relate linear amplification and acoustic parameters to changes in performance, differences in performance over hearing condition (and emotions) will be described in relation to the distance matrices and patterns of confusion from the acoustic analyses.

The dependent/outcome variable in all analyses will be raw accuracy. However, in addition, we will report Rosenthal's Proportion Index (PI, [41]) for each emotion expression of both stimulus types, to facilitate future comparison between studies.

## 3. Timeline

Ongoing—December 2021- Pilot study-Validation of recordings of emotion expressions. Analysis and selection of material for the presented study.

December 2021-November 2022: Data collection and acoustic analyses.

December 2022-March/April 2023: Analyses of results.

June 2023: Stage 2 report. Submission of article.

## Author Contributions

**Conceptualization:** Mattias Ekberg, Josefine Andin, Stefan Stenfelt, Örjan Dahlström.

**Data curation:** Mattias Ekberg, Örjan Dahlström.

**Formal analysis:** Mattias Ekberg, Örjan Dahlström.

**Investigation:** Mattias Ekberg.

**Methodology:** Mattias Ekberg, Josefine Andin, Stefan Stenfelt, Örjan Dahlström.

**Resources:** Stefan Stenfelt.

**Software:** Örjan Dahlström.

**Supervision:** Josefine Andin, Örjan Dahlström.

**Validation:** Mattias Ekberg.

**Visualization:** Mattias Ekberg.

**Writing – original draft:** Mattias Ekberg, Josefine Andin, Stefan Stenfelt, Örjan Dahlström.

**Writing – review & editing:** Mattias Ekberg, Josefine Andin, Stefan Stenfelt, Örjan Dahlström.

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
