## [Decision Letter · Decision Letter 0]

11 Aug 2021

PONE-D-21-13058

Effects of mild-to-moderate sensorineural hearing loss and signal amplification on vocal emotion recognition in middle-aged–older individuals.

PLOS ONE

Dear Dr. Ekberg,

Thank you for submitting your manuscript to PLOS ONE. After careful consideration, we feel that it has merit but does not fully meet PLOS ONE’s publication criteria as it currently stands. Therefore, we invite you to submit a revised version of the manuscript that addresses the points raised during the review process.

We look forward to receiving your revised manuscript.

Kind regards,

Qian-Jie Fu, Ph.D.

Academic Editor

PLOS ONE

1. Please ensure that your manuscript meets PLOS ONE's style requirements, including those for file naming. The PLOS ONE style templates can be found at https://journals.plos.org/plosone/s/file?id=wjVg/PLOSOne_formatting_sample_main_body.pdf and https://journals.plos.org/plosone/s/file?id=ba62/PLOSOne_formatting_sample_title_authors_affiliations.pdf.

2. Please include 'Registered Report Protocol' in the title of your manuscript.

Additional Editor Comments (if provided):

Reviewers' comments:

Reviewer's Responses to Questions

**Comments to the Author**

1. Does the manuscript provide a valid rationale for the proposed study, with clearly identified and justified research questions?

Reviewer #1: Partly

Reviewer #2: Partly

2. Is the protocol technically sound and planned in a manner that will lead to a meaningful outcome and allow testing the stated hypotheses?

Reviewer #1: Partly

Reviewer #2: Partly

3. Is the methodology feasible and described in sufficient detail to allow the work to be replicable?

Reviewer #1: No

Reviewer #2: Yes

4. Have the authors described where all data underlying the findings will be made available when the study is complete?

Reviewer #1: No

Reviewer #2: Yes

5. Is the manuscript presented in an intelligible fashion and written in standard English?

Reviewer #1: Yes

Reviewer #2: Yes

6. Review Comments to the Author

You may also provide optional suggestions and comments to authors that they might find helpful in planning their study.

Reviewer #1: Abstract

The authors’ rationale for the current study is that existing research cannot distinguish a specific emotion-perception deficit from a general prosody-perception deficit in listeners with hearing loss, because emotion identification of verbal materials depends on prosody perception. The authors plan to use both verbal and nonverbal emotional speech materials to examine if listeners with hearing loss have a true emotion-specific deficit, and not just a general prosody-related deficit.

The abstract could be edited to make the rationale clearer and more coherent. To improve the flow of the abstract, I suggest focusing on the specific- vs general-deficit issue before moving on to the topics of audibility and hearing aids, instead of introducing information about hearing aid studies and then switching back to the first issue. It is also not clear in the abstract how changing the audibility would help to answer the question of whether listeners have a specific or general deficit, or how examining “mix-ups” would be helpful.

Sentences that need editing include: “…deficits in individuals hearing loss also ha are not ameliorated…” and “…mix-ups between different vocal of individuals…”.

Introduction

The study aim of relating acoustic features to emotion recognition was mentioned in the “vocal emotion recognition” section and included in the Aims section, but was not described in the abstract.

“Patterns of confusion” is a slightly different concept than “differences in accuracy among emotions”. The authors should consider which is most relevant for their purposes and be consistent in their terminology, instead of using these terms interchangeably.

The authors could explain what they mean by “examining patterns of confusion [will lead to] deeper knowledge of emotion recognition” (end of Introduction); e.g., the relevant explanation at the end of the Aims section could be brought up earlier.

- missing ’s’ in ‘material’ at the end of the first paragraph

- pg 3 line 8: clarify what “in contrast” is contrasting

- pg 3 line 10: “however” implies the pattern for prosody and non-verbal vocalizations should be different, but they seem more alike than different

- proofread for extra punctuation, e.g., ‘;,’ at the end of page 3, or missing punctuation, e.g., missing period after ‘poorer pitch perception (18)’

Aims

- the phrase “for emotions expressed non-verbally when linear amplifications is not used” seems to be redundant, given that the first part of the sentence already both types of materials would be more poorly recognized regardless of amplification

- extra ’s’ in ‘amplifications’

Method - participants

Regarding the a priori power calculation, the meaning of the following statement is unclear: “we will be able to conduct separate analyses for different stimulus types and outcome measures”. I assume that it means that the same number of participants is appropriate for a 2 x 7 within-subjects design, for comparing amplified and non-amplified speech materials, and for comparing non-verbal vocalizations to sentences, but it would be clearer to say so explicitly.

Method - task and study design

The sentence materials are described in detail, but there is no description of what the “non-verbal vocalizations” consist of. I see that the “neutral” option is also excluded; is there no possibility of a “neutral” non-verbal vocalization?

What is the accuracy criterion for emotions to be recognized “well above chance” for pilot testing of the speech materials, and will it be in line with previous studies?

The authors should clarify which parts of the text describe pilot testing and which parts describe the procedures for the actual study, e.g., by using a separate sub-header.

For reader unfamiliar with this master hearing aid system, the authors could clarify if the “amplified” stimuli will be pre-processed and tailored to each participant’s audiogram before presentation in the session — is this the case?

If participants are presented with 8 options instead of 7, including the extra option of “other emotion”, wouldn’t this technically create 8 levels in the condition (instead of 7 as in the power calculation)? This also seems to add an extra complication to the calculation of chance levels for recognition, given that actors were directed to create only 7 emotions.

Method - analyses

A very brief description of GeMAPS would help readers to understand why the authors chose to use this set of speech measures.

How will the distance matrices of the acoustic measurements be integrated with the behavioral data (accuracy)?

Reviewer #2: Introduction

-page 2, first paragraph, they’ve been identified as an important question… why? Especially auditory rehabilitation (do you mean use of amplification devices? “auditory rehab” usually pertains to behavioural strategies).

-maybe in this first paragraph you want to discuss the number of individuals (esp older adults) with hearing loss, the prevalence of this condition in that population makes addressing issues related to age-related hearing loss a pressing concern. You actually don’t really talk about hearing loss in middle-aged/older individuals (your target population for the proposed study) at all, or why it is so important to study these people.

-page 3, first paragraph. There are a lot of data here about differences in recognition rates of emotions, and how they vary between emotional prosody and non-verbal vocalizations. Could be helpful to provide a table?

End of first paragraph, page 3, reword “why we…”

If less is known, is there any information that is known? Any previous studies on this?

Add a line about how hearing aids are most common intervention but still such a low uptake. Suggests that many people (esp. older adults) with hearing loss are not accessing amplification.

Top of page 4, Christensen and Goy papers, are these with older individuals experiencing hearing loss? Is there a confusion between age x emotion x hearing loss? Any expectations why aging might change ability to recognize emotion?

Method

2.2- can you break down into sub-heading first describing the pilot study, then another section describing the proposed full set of stimuli

2.3 Analyses- can you break down into behavioural data (participant responses) and acoustical analysis of your stimuli, starting with “for each recording…”

It would be helpful to describe where your data will be stored (institutional website?).

7. PLOS authors have the option to publish the peer review history of their article (what does this mean?). If published, this will include your full peer review and any attached files.

Reviewer #1: No

Reviewer #2: No

---

## [Author Response · Author response to Decision Letter 0]

22 Oct 2021

Reviewer #1

Abstract

Reviewer comment 1:1

The authors’ rationale for the current study is that existing research cannot distinguish a specific emotion-perception deficit from a general prosody-perception deficit in listeners with hearing loss, because emotion identification of verbal materials depends on prosody perception. The authors plan to use both verbal and nonverbal emotional speech materials to examine if listeners with hearing loss have a true emotion-specific deficit, and not just a general prosody-related deficit.

The abstract could be edited to make the rationale clearer and more coherent. To improve the flow of the abstract, I suggest focusing on the specific- vs general-deficit issue before moving on to the topics of audibility and hearing aids, instead of introducing information about hearing aid studies and then switching back to the first issue. It is also not clear in the abstract how changing the audibility would help to answer the question of whether listeners have a specific or general deficit, or how examining “mix-ups” would be helpful.

Sentences that need editing include: “…deficits in individuals hearing loss also ha are not ameliorated…” and “…mix-ups between different vocal of individuals…”.

Response 1:1

Thank you for the suggestions about the abstract, we have now rewritten the abstract to make the rationale clearer and more coherent. The new abstract now reads as follows: 

” Previous research has shown deficits in vocal emotion recognition in sub-populations of individuals with hearing loss, making this a high priority research topic. However, previous research has only examined vocal emotion recognition using verbal material, in which emotions are expressed through emotional prosody. There is evidence that older individuals with hearing loss suffer from deficits in general prosody recognition, not specific to emotional prosody. No study has examined the recognition of non-verbal vocalization, which constitutes another important source for the vocal communication of emotions. It might be the case that individuals with hearing loss have specific difficulties in recognizing emotions expressed through prosody in speech, but not non-verbal vocalizations. We aim to examine whether vocal emotion recognition difficulties in middle- aged-to older individuals with sensorineural mild-moderate hearing loss are better explained by deficits in vocal emotion recognition specifically, or deficits in prosody recognition generally by including both sentences and non-verbal expressions. Furthermore a,, some of the studies which have concluded that individuals with mild-moderate hearing loss have deficits in vocal emotion recognition ability have also found that the use of hearing aids does not improve recognition accuracy in this group. We aim to examine the effects of linear amplification and audibility on the recognition of different emotions expressed both verbally and non-verbally. Besides examining accuracy for different emotions we will also look at patterns of confusion (which specific emotions are mistaken for other specific emotion and at which rates) during both amplified and non-amplified listening, and we will analyze all material acoustically and relate the acoustic content to performance. Together these analyses will provide clues to effects of amplification on the perception of different emotions. For these purposes, a total of 70 middle-aged-older individuals, half with mild-moderate hearing loss and half with normal hearing will perform a computerized forced-choice vocal emotion recognition task with and without amplification.”

Introduction

Reviewer comment 1:2

The study aim of relating acoustic features to emotion recognition was mentioned in the “vocal emotion recognition” section and included in the Aims section, but was not described in the abstract.

Response 1:2 

Thank you for noticing and pointing this out. This aim is now included in the abstract. See Response 1:1. 

Reviewer comment 1:3

“Patterns of confusion” is a slightly different concept than “differences in accuracy among emotions”. The authors should consider which is most relevant for their purposes and be consistent in their terminology, instead of using these terms interchangeably.

Response 1:3

We agree that there is a difference between the two terms. We distinguish between accuracy for different emotions and patterns of confusion throughout the manuscript. 

Reviewer comment 1:4

The authors could explain what they mean by “examining patterns of confusion [will lead to] deeper knowledge of emotion recognition” (end of Introduction); e.g., the relevant explanation at the end of the Aims section could be brought up earlier.

Response 1:4

Thank you for pointing out the need for clarification here. In our revised aims section we now write (end of pg.4- pg.5): 

“More specifically, by examining accuracy for different emotions (what emotions are easiest to accurately identify) and patterns of confusion (which emotions are mixed up when inaccurately identified), for individuals with normal hearing and for individuals with hearing loss (listening with and without linear amplification), and by comparing performance with acoustic analysis of sentences and non-verbal expressions with different emotional emphasis, the aims are to examine:

• Which acoustic parameters are important for detection of and discrimination between different emotions (by examination of performance of normal hearing participants)?

• How the effect of hearing loss affects that pattern (by comparing performance of participants with and without hearing loss)?

• How that pattern is affected by linear amplification (by examination of performance of participants with hearing loss using linear amplification)? “

We also introduce the topic of confusion between different emotions in individuals with hearing loss in the section “Vocal emotion recognition under mild-to-moderate hearing loss” in the introduction:

“In individuals with hearing loss, confusions between different vocal emotions appear to be much more common in comparison to normal hearing individuals (31)” (pg. 4, §3)

Reviewer comment 1:5

- missing ’s’ in ‘material’ at the end of the first paragraph

Response 1:5

Thanks for noticing this error. The ’s’ has been added

Reviewer comment 1:6

- pg 3 line 8: clarify what “in contrast” is contrasting

Response 1:6

Thanks for pointing this inconsistency out. This part of the text has been removed from the current manuscript.

Reviewer comment 1:7

- pg 3 line 10: “however” implies the pattern for prosody and non-verbal vocalizations should be different, but they seem more alike than different

Response 1:7

Thank you for this accurate observation. The wording has been changed to:

”However, with regard to emotional prosody specifically, Scherer (5) and Castiajo and Pinheiro (16) argue that anger and sadness are more accurately recognized compared to fear, surprise, and happiness.”

Reviewer comment 1:8

- proofread for extra punctuation, e.g., ‘;,’ at the end of page 3, or missing punctuation, e.g., missing period after ‘poorer pitch perception (18)’

Response 1:8

Thank you for noticing. Missing period after ’poorer pitch perception’ has been added. The extra semicolon on pg. 3 has been removed. 

Aims

Reviewer comment 1:9

- the phrase “for emotions expressed non-verbally when linear amplifications is not used” seems to be redundant, given that the first part of the sentence already both types of materials would be more poorly recognized regardless of amplification

Response 1:9

Thank you for noticing this mistake. The preceding sentence refers to verbal expressions and this to non-verbal. We have rewritten the sentences and clarified it:

“...we predict that: 

1. individuals with hearing loss will have poorer recognition compared to the normal hearing group for emotions expressed verbally, regardless of acoustic features and regardless of the use of linear amplification, and for emotions expressed non-verbally when linear amplifications is not used”

Reviewer comment 1:10

- extra ’s’ in ‘amplifications’

Response 1:10

Thank you for noticing. An extra s has been added (pg.5, top of page and lines below).

Method - participants

Reviewer comment 1:11

Regarding the a priori power calculation, the meaning of the following statement is unclear: “we will be able to conduct separate analyses for different stimulus types and outcome measures”. I assume that it means that the same number of participants is appropriate for a 2 x 7 within-subjects design, for comparing amplified and non-amplified speech materials, and for comparing non-verbal vocalizations to sentences, but it would be clearer to say so explicitly.

Response 1:11

Thank you. We agree that this section was unclear. This text has been changed to: 

“Using a 2 x 6 mixed design (group x emotion), 80% power, 5% significance level, correlation between repeated measures of 0.5 and with N=56 participants (a reasonable number of participants from a recruitment point of view), we will be able to detect an effect as small as η2=.02 (f=0.14); a next to small effect size. Power analysis was performed using G*Power version 3.1.9.7 (32).” (pg. 5, §3)

As is discussed in the analyses section, several of the planned analyses will be 2x6 mixed ANOVAs, but not all. However, we think this is the most demanding analysis in terms of number of participants, so we based our estimation on this, along with consideration regarding how many participants we will realistically will be able to recruit. 

Method - task and study design

Reviewer comment 1:12

The sentence materials are described in detail, but there is no description of what the “non-verbal vocalizations” consist of. I see that the “neutral” option is also excluded; is there no possibility of a “neutral” non-verbal vocalization?

Response 1:12

Thank you for noticing this omission. We have added the following description in the Task and study design section ( pg. 5, §2)

”For the non-verbal expressions the actors were instructed to imagine themselves experiencing the different emotions and to make expressions with sounds, without language which match those emotions. This entails some variation of the specific sounds expressed for specific emotions by different participants. The sounds which were perceived as most clearly expressing a given emotion were selected by the authors, and some were slightly edited (omitting pauses) such that all sounds are between 1-2 seconds long.”

With regard to the exclusion of neutral, we believe that such expressions would be very difficult to achieve, if they are achievable. We have not found any previous study including neutral for non-verbal expressions. 

Reviewer comment 1:13

What is the accuracy criterion for emotions to be recognized “well above chance” for pilot testing of the speech materials, and will it be in line with previous studies?

Response 1:13

There does not seem to be a generally accepted criterion for validation at the level of individual stimuli. The commonly used criterion of above-chance accuracy seems to be used at a category level such as overall accuracy for expressions of a particular category. We have simply chosen stimuli which are accurately classified by a majority of participants (>50%). The new description under the subheading Validation of stimulus material reads: 

“Only sentences and non-verbal expressions which a majority of participants (>50%) classify as expressing the emotion intended by the actor will be included in the experiment” (pg.6, §3)

Reviewer comment 1:14

The authors should clarify which parts of the text describe pilot testing and which parts describe the procedures for the actual study, e.g., by using a separate sub-header.

Response 1:14

Thank you for this suggestion. The pilot study is now described under the new heading “Validation of stimulus material”. (pg. 6, §3)

Reviewer comment 1:15

For reader unfamiliar with this master hearing aid system, the authors could clarify if the “amplified” stimuli will be pre-processed and tailored to each participant’s audiogram before presentation in the session — is this the case?

Response 1:14

The master hearing aid system will process the auditory stimuli tailored to the participant’s audiogram during presentation, such that the amplified stimuli will be presented via head- phones to the participants, who at the time will not wear their own hearing aids. We have added the following description- pg. 6, §4 : 

”.. with linear amplification using the Cambridge formula for linear hearing aids (37) in which each stimulus is tailored to each participants audiogram during presentation”

Reviewer comment 1:15

If participants are presented with 8 options instead of 7, including the extra option of “other emotion”, wouldn’t this technically create 8 levels in the condition (instead of 7 as in the power calculation)? This also seems to add an extra complication to the calculation of chance levels for recognition, given that actors were directed to create only 7 emotions

Response 1:15

The option of ’other emotion’ is only included in the pilot study/validation. We agree that adding the option of “other emotion” does add complication to the calculation of chance levels, although it seems to be standard to include such an option in validation studies to reduce bias. Our new criterion as described however does not take into account the number of options. 

Method - analyses 

Reviewer comment 1:16

A very brief description of GeMAPS would help readers to understand why the authors chose to use this set of speech measures.

Response 1:16

The following description has been added- pg. 7, §2: ” The GeMAPS consist of a set of acoustic parameters which have been proposed as a standard for different areas of automatic voice analysis, including the analysis of vocal emotions. The parameters of GeMAPS have been shown to be of value for analyzing emotions in speech in previous research (40) and are described in table 1.”

Reviewer comment 1:17

How will the distance matrices of the acoustic measurements be integrated with the behavioral data (accuracy)?

Response 1:17

We will discuss differences in accuracy and patterns of confusion in relation to descriptive graphical plots of distance matrices for the central acoustic parameters. We have added the following under Behavioral analyses, pg. 8, §1 of the subsection: 

” To relate linear amplification and acoustic parameters to changes in performance, differences in performance over hearing condition (and emotions) will be described in relation to the distance matrices and patterns of confusion from the acoustic analyses”

Reviewer #2:

Reviewer comment 2:1

Introduction

-page 2, first paragraph, they’ve been identified as an important question… why? Especially auditory rehabilitation (do you mean use of amplification devices? “auditory rehab” usually pertains to behavioural strategies).

Response 2:1

Picou et al (2018) identified that the subject of emotion recognition in hearing loss has been slightly neglected in audiology. They highlight the importance of emotion recognition due to it’s important for interpersonal interaction and social functioning and discuss a number of topics for which there is a need for research, including emotion recognition (more knowledge of how it is affected by hearing loss), the effects of amplification (long term and short) and different rehabilitation strategies (what works and how) with regard to vocal emotion recognition. 

With “auditory rehabilitation” we initially meant all types of interventions including amplification. We now realize however that auditory rehabilitation (referring primarily to behavioral intervantions should be distinguished from amplification (through devices such as hearing aids and CIs). Therefore, we now separate between the two in the text. 

We have added the following to pg. 2, §1: 

”Because of the importance of emotion recognition for interpersonal interaction and, thus, indirectly to measurements of well-being such as perceived quality of life and the presence/absence of depression, the effects of hearing loss, amplification, and auditory rehabilitation on the recognition and experience of vocal emotions, have been identified as important research topics (3)”

Reviewer comment 2:2

-maybe in this first paragraph you want to discuss the number of individuals (esp older adults) with hearing loss, the prevalence of this condition in that population makes addressing issues related to age-related hearing loss a pressing concern. You actually don’t really talk about hearing loss in middle-aged/older individuals (your target population for the proposed study) at all, or why it is so important to study these people.

Response 2:2

Thank you for this suggestion. We agree that this will be a good intro and have therefore added such information to the first paragraph of the Introduction. We have here also clarified the importance of investigating hearing loss in middle-aged individuals. As a consequence, the part introducing the meaning of emotions has been moved down under the subheading ’vocal emotion recognition’. The first part of the introduction now reads: 

”Hearing loss is among the leading causes of disability globally with an increasing prevalence with age. It leads to difficulties in communication which can contribute to social isolation and diminished well-being (1). In Sweden, it has been estimated that approximately 18% of the population have a hearing loss. While the prevalence is comparatively much higher in the oldest age groups (44.6 % in the age group of 75-84 and 56.5% in the age group of 85+ years) the prevalence in the age groups of 55-64 and 65-74 (middle-age-to-old) is also relatively high (24.2 and 32.6%) (2). This shows that hearing loss is not only a concern for the oldest in the population. Because of the importance of emotion recognition for interpersonal interaction and, thus, indirectly to measurements of well-being such as perceived quality of life and the presence/absence of depression, the effects of hearing loss, amplification, and auditory rehabilitation on the recognition and experience of vocal emotions, have been identified as important research topics (3).”

Reviewer comment 2:3

-page 3, first paragraph. There are a lot of data here about differences in recognition rates of emotions, and how they vary between emotional prosody and non-verbal vocalizations. Could be helpful to provide a table?

Response 2:3

We agree that this paragraph was dense and difficult to follow. We have considered providing a table but decided to keep the information as text. However, we have revised this section to make it clearer, as follows: 

”Research on how well different emotions of both stimulus types are recognized compared to other emotions has so far not yielded consistent results (see for example 4 and 15). However, with regard to emotional prosody specifically, Scherer and Castiajo and Pinheiro argue that anger and sadness are more accurately recognized compared to fear, surprise, and happiness (4, 15) (pg. 3, §.§, lines 12-16)

Reviewer comment 2:4

End of first paragraph, page 3, reword “why we…”

Response 2:4

This part has now been rewritten and reads as follows: 

“Therefore, in the present study, recognition of non-verbal expressions as well as emotional prosody will be investigated.” (pg.3, last two lines of §1)

Reviewer comment 2:5

If less is known, is there any information that is known? Any previous studies on this?

Response 2:6

We have not found any study using non-verbal expressions with older individuals with hearing loss. We have added the following, with the addition of a new reference (nr 18) on pg. 3, end of §1, lines 29-32: 

” The ability to accurately recognize non-verbal emotion expressions declines with aging (18), as does the ability to recognize emotional prosody in speech (19). Amorim et al. suggest that this decline happens due to age-related changes in brain regions which are involved in the processing of emotional cues (18). Hearing loss and aging both independently and conjointly contribute to a diminished ability to accurately recognize speech embedded vocal emotion expressions generally (19). To our knowledge, however, no previous study has examined the effects of mild to -moderate sensorineural hearing loss on non-verbal emotion expressions..” 

Reviewer comment 2:7

Add a line about how hearing aids are most common intervention but still such a low uptake. Suggests that many people (esp. older adults) with hearing loss are not accessing amplification.

Response 2:7

We have added the following at -pg. 4, §2, with an added new reference: 

”However, many individuals with hearing loss do not have access to or do not use hearing aids (25)”.

Reviewer comment 2:8

Top of page 4, Christensen and Goy papers, are these with older individuals experiencing hearing loss? Is there a confusion between age x emotion x hearing loss? Any expectations why aging might change ability to recognize emotion?

Response 2:8

The Goy et al. Paper examines the correlations between age and emotion identification and between PTA and emotion identification independently. Both age and PTA correlate significantly with emotion identification. In Christensen et al. participants are divided into different age groups. They found that aging does impact vocal emotion recognition negatively and that aging and hearing loss independently have negative effects on vocal emotion recognition accuracy, however the effects of hearing loss and old age are also additive. The suggested reasons for the age-related decline include both biological and social factors, we have added a brief discussion of one such proposed explanation, along with a very brief discussion of the effects of hearing loss and aging (see response 2:6): 

”The ability to accurately recognize non-verbal emotion expressions declines with aging (18), as does the ability to recognize emotional prosody in speech (19). Amorim et al. suggest that this decline happens due to age-related changes in brain regions which are involved in the processing of emotional cues (18). Hearing loss and aging both independently and conjointly contribute to a diminished ability to accurately recognize speech embedded vocal emotion expressions generally (19).”

Method

Reviewer comment 2:9

2.2- can you break down into sub-heading first describing the pilot study, then another section describing the proposed full set of stimuli

Response 2:9

The pilot study, stimuli material and the experimental study design are now presented under separate sub-headers (see pg. 6).

Reviewer comment 2:10

2.3 Analyses- can you break down into behavioural data (participant responses) and acoustical analysis of your stimuli, starting with “for each recording…”

Response 2:10

Analyses have been broken down into ’behavioral data’ and ’acoustic analyses’ under separate sub-headings (see pg. 7 for Acoustic analyses and pg.8 for Behavioral data)

Reviewer comment 2:11

It would be helpful to describe where your data will be stored (institutional website?).

Response 2:11

We intend to store data on the Open Science Framework (OSF) website. However, we have not yet created a link for this as the study has not started.

---

## [Decision Letter · Decision Letter 1]

1 Dec 2021

Effects of mild-to-moderate sensorineural hearing loss and signal amplification on vocal emotion recognition in middle-aged–older individuals.

PONE-D-21-13058R1

Dear Dr. Ekberg,

We’re pleased to inform you that your manuscript has been judged scientifically suitable for publication and will be formally accepted for publication once it meets all outstanding technical requirements.

Kind regards,

Qian-Jie Fu, Ph.D.

Academic Editor

PLOS ONE

Additional Editor Comments (optional):

Reviewers' comments:

Reviewer's Responses to Questions

**Comments to the Author**

1. Does the manuscript provide a valid rationale for the proposed study, with clearly identified and justified research questions?

Reviewer #1: Yes

2. Is the protocol technically sound and planned in a manner that will lead to a meaningful outcome and allow testing the stated hypotheses?

Reviewer #1: Yes

3. Is the methodology feasible and described in sufficient detail to allow the work to be replicable?

Reviewer #1: Yes

4. Have the authors described where all data underlying the findings will be made available when the study is complete?

Reviewer #1: Yes

5. Is the manuscript presented in an intelligible fashion and written in standard English?

Reviewer #1: Yes

6. Review Comments to the Author

You may also provide optional suggestions and comments to authors that they might find helpful in planning their study.

Reviewer #1: The authors' revisions have strengthened the rationale and improved the clarity of the manuscript overall; I have no further suggestions.

There is a minor typo in the abstract, "Furthermore a,," and a missing period before "Preliminary results" on p6.

7. PLOS authors have the option to publish the peer review history of their article (what does this mean?). If published, this will include your full peer review and any attached files.

Reviewer #1: No